# Impact of the COVID-19 Pandemic on Melanoma Diagnosis: Increased Breslow Thickness in Primary Melanomas—A Single Center Experience

**DOI:** 10.3390/ijerph192416806

**Published:** 2022-12-14

**Authors:** Jelena Jeremić, Branko Suđecki, Kristina Radenović, Jovan Mihaljević, Ivan Radosavljević, Milan Jovanović, Nataša Milić, Vedrana Pavlović, Dimitrije Brašanac, Marko Jović

**Affiliations:** 1Department of Plastic Surgery, Clinic for Burns, Plastic and Reconstructive Surgery, University Clinical Center of Serbia, 11000 Belgrade, Serbia; 2Faculty of Medicine, University of Belgrade, 11000 Belgrade, Serbia; 3Institute for Medical Statistics and Informatics, Faculty of Medicine, University of Belgrade, 11000 Belgrade, Serbia; 4Institute of Pathology, Faculty of Medicine, University of Belgrade, 11000 Belgrade, Serbia

**Keywords:** COVID-19, restrictions, melanoma, Breslow thickness, diagnostic delay

## Abstract

Early melanoma diagnosis plays a key role in ensuring best prognosis with good survival rates. The ongoing global COVID-19 pandemic has greatly impacted global and national healthcare systems, thus making it a real challenge. The aim of this study was to evaluate the impact of the pandemic on diagnostic delay in melanoma patients in Serbia. In this retrospective study, we included patients treated at the university hospital in Serbia’s capitol over a period of five years and three months. We compared the prepandemic (01/JAN/17-14/MAR/20) and pandemic periods (15/MAR/20-31/MAR/22) by evaluating patient demographic data, melanoma subtype, Breslow thickness, Clark level, ulceration status, mitotic index rate and pT staging. We observed a significant reduction in the number of diagnosed patients (86.3 vs. 13.7%; *p* = 0.036), with melanomas having an increased median Breslow thickness (1.80 vs. 3.00; *p* = 0.010), a higher percentage of Clark IV–V level lesions (44.0% vs. 63.0%; *p* = 0.009), an increase in median mitotic index rate (2 vs. 5; *p* < 0.001) and a trend of increase in lesions thicker than 2 mm (37.8% vs. 53.7%; *p* = 0.026). We believe that this study can be a useful scenario guide for future similar events, highlighting the importance of preventive measures and timely diagnosis for the best patient outcomes.

## 1. Introduction

Cutaneous melanoma is considered to be one of the most lethal forms of skin cancer and accounts for the majority of skin cancer deaths [1]. Over the past few decades, the incidence of melanoma has risen dramatically worldwide, especially in regions with fair-skinned populations [1,2,3]. According to Okhovat et al., it is currently increasing faster than any other preventable cancer in the United States [4]. While in many European populations, such as the United Kingdom and the Netherlands, incidence rates have increased annually by 4 to 6% in recent decades, the annual increase in incidence in Australia and New Zealand seems to have leveled flat since 1995 [5]. A higher Breslow index and the presence of ulcerations indicate a more aggressive form of melanoma with a higher risk of local and distant metastases and a generally worse prognosis [6]. Melanoma tumor thickness is directly correlated to its disease-specific 10-year survival rate [7]. Therefore, an early diagnosis is important for the successful treatment of the disease and keeping mortality at a low level [1,2,8]. An essential part of early diagnosis is screening programs [2]. A rise in the implementation of these programs as well as the higher accessibility of dermoscopic exams over the past few decades has resulted in a higher percentage of melanomas being discovered in the earliest stages of the disease, thus lowering the number of advanced-staged lesions [9]. Melanomas detected by clinicians during skin examinations are more likely to be thinner than the melanomas that patients detect via self-examination [10].

The ongoing pandemic, caused by the SARS-CoV-2 virus, a new highly transmissible variant of coronavirus causing an acute respiratory disease called COVID-19, proved to be a significant challenge for public health. The virus was first detected in Wuhan (China) in late December 2019 but has since then quickly spread globally [11]. On 6th March 2020, the first patient in Serbia was diagnosed, which marked the beginning of our own COVID-19 public health crisis which has, up to this day, claimed more than 17,000 lives [12]. In 2020, COVID-19 was the second leading cause of death among the male residents of Belgrade, while in females, it was the third leading cause of mortality [13]. The first response in most countries was to contain the spread of the virus through immediate case detection and isolation, rigorous close-contact tracing with mandatory quarantine, and the application of strict limitations to people’s movements and activities through mandatory curfews [14,15,16]. In Serbia, this mandatory curfew lasted from 15th March until 6th May 2020. During this time, major nationwide restrictions on social and public life were implemented to contain the pandemic [17]. The pandemic has dramatically changed triage, diagnosing, and treatment procedures as well as workflows in hospitals, both nationally and globally [18]. This strategy has led Europe and the rest of the world to reduce elective hospital activities for nonurgent and non-COVID-related cases. Specialized outpatient clinics were forced to comply with a marked reduction in face-to-face consultations [19]. Elective surgical procedures were delayed and/or canceled by many. This was especially notable during the lockdown period [20]. Cancer screenings and even the surgical management of cancer patients were also greatly delayed. The goal of these measures was to avoid unnecessary exposure in high-risk environments through crowding in waiting rooms and wards [21]. At the same time, most hospitals were advised to reduce elective procedures and to prepare for the emerging number of COVID-19 patient admissions [22]. Many patients canceled their scheduled visits, some because of being in quarantine and others out of a growing fear of contagion [23]. Material and human resources were in great measure transferred to newly formed intensive care units to help manage the overwhelming rush of severe forms of COVID-19 being admitted every day [22,24]. 

In addition to the devastating consequences for patients, delays in melanoma treatment can have a profound impact on the economic burden of this disease, because advanced forms have higher healthcare costs than early-stage melanomas [25]. These pandemic-associated disruptions in cancer care raised worldwide concerns about delays in early-on melanoma diagnosis being associated with an increase in morbidity and mortality [26]. Since the beginning of the pandemic, several studies have reported a significant reduction in melanoma patient diagnoses [27,28,29]. In addition, it was reported that newly discovered melanomas were thicker during the pandemic period in comparison to the prepandemic era [29,30,31,32,33,34,35,36,37]. Moreover, a study by Lallas et al. reported a significantly higher-than-expected percentage of newly diagnosed melanomas in stages IIc, III, and IV after the 2020 lockdown when compared to previous years [38]. In contrast to the former findings, multiple studies found no pandemic-related impact on melanoma tumor depth [27,39,40,41,42]. 

To the best of our knowledge, to this date, no studies in Serbia have been performed to demonstrate the impact of the COVID-19 pandemic on the diagnosis and treatment of melanoma patients. This study aimed to identify the pandemic’s impact on Serbia’s national healthcare system, focusing on diagnostic delays in melanoma patients. Our hypothesis was that for all the above-mentioned reasons, patients treated during the pandemic period would have thicker, more advanced lesions when compared to the prepandemic group.

## 2. Materials and Methods

### 2.1. Study Design and Data Extraction

This retrospective study was conducted at the Clinic for Burns, Plastic and Reconstructive Surgery of the University Clinical Center of Serbia, whose area of influence covers the whole city of Belgrade and its 1,166,800 inhabitants, with a population density of 3241 inhab./km^2^. The clinic is also a tertiary national referral center for skin cancer patients. We included a sample of subsequent melanoma patients treated at our clinic between 1 January 2017 and 31 March 2022. Treated patients were partly referred by primary healthcare centers in Belgrade, while the rest were referrals from secondary and tertiary institutions in other parts of Serbia. Patients who had previously been diagnosed and treated at another facility and then referred to us for additional sentinel lymph node biopsy were not included in this study. All patients received a full preoperative examination and were later treated according to the current American Joint Committee on Cancer (AJCC) 2017 8th edition as well as National Comprehensive Cancer Network (NCCN) guidelines. All specimens were promptly sent and analyzed at the Institute for Pathology of the Medical Faculty in Belgrade by a pathologist experienced and versed in analyzing melanocytic lesions. After ethics committee approval, the data were extracted from patient files and the clinic’s pathology results archive. We obtained general patient demographic data (age and sex), date of biopsy, primary lesion localization, and melanoma-specific characteristics (subtype, Breslow thickness, Clark level, mitotic index rate, and ulceration status). In order to determine the impact of the COVID-19 pandemic on melanoma characteristics, all patients were divided into two cohorts: prepandemic (biopsies taken between 1 January 2017 and 14 March 2020) and pandemic (biopsies taken between 15 March 2020 and 31 March 2022). The pandemic cohort timeframe was selected following the beginning of the government-issued lockdown, up until all previously set restrictions and regulations were lifted. In terms of age at the moment of biopsy, all patients were divided into four categories: (a) ≤40, (b) 41–60, (c) 61–80, and (d) >80 years of age. According to the primary site distribution, all melanomas were divided into 4 groups: (a) head and neck, (b) torso, (c) upper extremities, (d) lower extremities. Melanoma-specific tumor characteristics were then recorded and analyzed. All samples were histologically divided into 4 major groups: (a) superficial spreading (SSM), (b) nodular (NOD), (c) lentigo maligna (LMM), (d) other (acral lentiginous (ALM), nevoid, spitzoid, dermal, desmoplastic, meltump, malignant blue nevus, polipoid, and Reed nevus like melanoma). Breslow thickness was, according to the AJCC 2017 8th edition guidelines, divided into four categories: <1 mm, 1.01–2 mm, 2.01–4 mm, and >4 mm; Clark depth was categorized into five levels (1–5). The mitotic index rate was calculated using the hot spot method/mm^2^, while ulceration status was marked as present vs. absent. The staging was conducted according to tumor depth (in situ, pT1–T4), based on the melanoma staging criteria of the AJCC 2017 8th edition.

### 2.2. Data Analysis

Categorical variables are presented as absolute numbers with percentages. Numeric variables were presented as means with standard deviations or medians with 25th–75th percentile according to data distribution. Differences between patients treated before and during the COVID-19 pandemic were analyzed by Student’s *t*-test or a Mann–Whitney test, as appropriate. Associations between categorical data were evaluated using the Pearson chi-square test. Violin plots were used to present differences in Breslow score in patients treated before and during the COVID-19 pandemic [43]. The level of significance was set at 0.05. Statistical analysis was performed using the IBM SPSS 21 (Chicago, IL, USA, 2012) package.

## 3. Results

### 3.1. Patient Characteristics before and during the Pandemic

During a study period of 5 years, a total of 393 patients with melanoma were diagnosed. There were 73 diagnosed patients in 2017, 116 in 2018, 114 in 2019, 33 in 2020, 42 in 2021 (Figure 1), and 15 during the first three months of 2022. Out of all examined lesions, 311 (79.1%) were obtained prior to and 82 (20.9%) during the pandemic period. This equates to 101.0 ± 24.3 diagnosed cases per year prior and 37.5 ± 6.4 during the pandemic (Table 1). When compared to previous years, a 58.2% reduction in annual melanoma diagnoses was found (*p* = 0.036). 

The studied samples were comprised of 219 (55.7%) males and 174 (44.3%) females. Overall, the majority of studied patients were in the 60–80 age group (Table 1). The median patient age in the prepandemic group was 68.0 (range 19.0–97.0), while in the pandemic group, it was 71.0 (range 28.0–88.0). There were no significant differences in sex and age of the patients before and after the onset of the pandemic (*p* = 0.978, *p* = 0.852, respectively) (Table 1). 

### 3.2. Body Distribution of Melanomas before and during the Pandemic

The most frequent anatomic site of melanoma was the trunk (39.2%), followed by the head and neck (20.6%) as well as lower (20.9%) and upper extremities region localization (19.3%) (Table 1). Although there was no significant difference in the individual body distribution of melanomas between the cohorts, it is notable that the pandemic cohort had an increased percentage of head, neck, and torso melanomas with a decreased percentage of upper and lower extremity melanomas (74.1% vs. 25.9%; *p* = 0.021) (Table 1). These lesions in both cohorts occurred more often in men than in women (69.4% vs. 47.7%, *p* < 0.001). 

### 3.3. Melanoma-Specific Histologic Characteristics before and during the Pandemic

Regarding the histological characteristics of melanoma (Table 1), SSM was the most commonly obtained histologic subtype in both groups (65.6%). No significant differences regarding histologic subtypes between the two time periods were observed (*p* = 0.137). Although no significant differences regarding histologic subtypes between the two time periods were observed (*p* = 0.137), we noted 17.4% vs. 25.9% cases of nodular subtype in the two groups, respectively (Table 1). In the prepandemic cohort, 2.9% of patients had ALM, while in the pandemic cohort, we obtained no such lesions. 

Melanomas excised during the pandemic period showed a trend toward a higher Breslow thickness median (1.80 vs. 3.00; *p* = 0.010) when compared to the prepandemic cohort (Figure 2). Lesions obtained during the pandemic were notably thicker in women (median 1.20 vs. 2.40; *p* = 0.044) as well as in patients older than 40 (1.90 vs. 3.50; *p* = 0.007). 

The pandemic cohort more frequently presented with melanoma lesions thicker than 2 mm compared to the prepandemic, which coincided with a higher proportion of pT3-T4 cases (37.8% vs. 53.7%; *p* = 0.026) (Figure 3). Other differences in pT staging distribution regarding patient sex and age during the pandemic were not observed. No change in the percentage of in situ and invasive forms between the two groups was observed (*p* = 0.521) with the majority of samples in both cohorts being invasive forms (83.2% vs. 79.6%). 

Individual Clark level distributions showed statistically significant changes in distribution pre and during the pandemic (*p* = 0.029), with the majority in both groups being a Clark IV lesion (38.1% and 55.6%) (Table 1). Upon further statistical analysis, we noticed a significant increase in the percentage of combined Clark IV–V lesions between the two cohorts (44.0% vs. 63.0%; *p* = 0.009). 

There was a change in the mitotic index rate in the pandemic cohort. The median mitotic index rate of melanomas included before the pandemic was 2, which was a value that increased to 5 in the subsequent pandemic time (*p* < 0.001) (Table 1). Prior to the pandemic, 33.7% of melanomas were ulcerated when compared to 44.2% in the pandemic group, although this difference did not reach statistical significance (*p* = 0.179) (Table 1). 

## 4. Discussion

### 4.1. Main Findings

This study accentuates the importance of early melanoma diagnosis as well as the role of Breslow thickness as an important prognostic factor in further melanoma management. Major changes happened during the pandemic, shifting most of the available resources toward managing COVID patients. This has led to a significant diagnostic delay in melanoma patients, which could have a great impact on their overall prognosis and long-term outcome. 

### 4.2. Impact of the COVID-19 Pandemic on Patient Referrals and the Number of Diagnosed Cases

In late 2019, a new variant of Coronavirus known as SARS-CoV-2 emerged, prompting World Health Organization (WHO) to declare a global pandemic on 11 March 2020 [11]. This pandemic has shaken global healthcare systems, the economy, and the general well-being of the population worldwide. Depending on its capacity, every nation has since then implemented different measures dedicated to stopping the spread of the disease as well as managing an exponentially growing number of newly diagnosed patients. Many studies showed that these restrictions have led to a drastic drop in the number of newly diagnosed skin tumors, some even by 60% [30,38,44]. Out of all patients with malignancies, skin tumor patients have been the ones with the most missed appointments, with a 56.7% decrease in diagnoses in 2020 when compared to the previous year [45]. The European Academy of Dermatology and Venerology (EADV) recommended that elective skin cancer screenings for individuals with increased melanoma risk be extended by a maximum of 2–3 months during the pandemic [46]. Although updated recommendations and guidance were issued for the surgical treatment of melanoma during the pandemic, they could not have assumed patient preferences or access for initial lesion evaluation in regional offices that were forced to close [47]. The International Dermoscopy Society (IDS) reported that since the beginning of the pandemic, there has been a 75% reduction in the daily work activity of its members during the lockdown period, with more than half of them diagnosing practically no melanomas in those months [48]. In a survey conducted by the Global Coalition for Melanoma Patient Advocacy, it was reported that 21% of melanoma cases went undiagnosed throughout the COVID-19 pandemic in 2020, which resulted in about 60,000 melanomas worldwide [49]. The Belgian Cancer Registry has reported that in comparison to the previous year, there was a 44% decrease in invasive cancer diagnoses through April 2020. They reported that the largest decline was in the diagnosis of non-melanoma skin cancer and melanoma, especially in the older population [50]. 

Following most countries, Serbia’s national healthcare facilities have quickly shifted most of their capacities and resources toward managing COVID-19 patients, which has in turn led to the majority of elective procedures being delayed or even canceled. In correlation with those findings, our study has also shown a significant drop in melanoma incidence during 2020 and 2021 when compared to previous years. This was time-related to the pandemic break-outs as well as state-implemented restrictions. Even before the pandemic, there have always been usual fluctuations regarding the frequency of patient consultations. The reasons behind them were usually the state holidays, seasonal weather changes (higher or lower temperatures making appointments more difficult on the elderly population), lack of hospital personnel during vacation times, etc. [51,52,53,54]. However, these usual fluctuations could not cause such a major gap in case numbers seen since the beginning of the pandemic in our country. 

Our results are in accordance with other studies, which have already reported the impacts of the COVID-19 pandemic on patient consultation and admission delays [55,56]. As expected, heavily struck regions, such as north Italy and Spain, have reported a significant drop in the number of melanoma diagnoses [33,57]. Ricci et al. reported that the number of newly diagnosed melanomas has dropped from 2.3 cases per day (CPD) prior to 0.6 CPD during the pandemic’s peak, with an increase to 1.3 CPD following the lifting of restrictions [30]. 

Contrary to these results, some authors, such as Kostner et al., have described an unchanged or even increased number of visits during and post-lockdown. This can be influenced by state regulations, restriction levels as well as the healthcare resource capacities of different regions [35]. Schauer et al. reported an increase in early-stage melanoma diagnoses during the lockdown period, which can be explained by the fact that screening programs were still available despite the government restrictions [58]. 

This accentuates the necessity and importance of screening programs’ availability. However, there is no doubt that the state-implemented restrictions were not the only reason for the decrease in patient visits. The pandemic has been heavily covered by the media, showing graphic pictures of healthcare facilities being turned into COVID wards, overwhelmed staff wearing extensive protective gear, warnings about the risk of contagion within hospital grounds, etc. [55]. Aside from that, many patients became fearful and anxious about contagion, whether from personal close encounters or hospital visits [59]. 

### 4.3. Melanoma-Specific Characteristics during the Pandemic

Regarding histologic subtypes, we observed no cases of ALM during the pandemic period. This could be due to the fact that acral regions, such as soles, are hard to reach and often unavailable to self-examination and in most cases can be only diagnosed with certainty by dermoscopic examinations, which were widely unavailable during the pandemic time. Although the Breslow thickness, ulceration status, and mitosis index rate have been described as the most important prognostic factors [6], recent studies by Nagore et al. and Gualdi et al. have shown that the time lapse between the first signs of a lesion and definitive diagnosis also plays an important role in the further management and prognosis of melanoma patients [60,61]. Specifically, the doubling time for melanoma is estimated to be around 94 days, as compared to 241 days for invasive breast cancer, 440 days for 600 days for lung adenocarcinoma, and 936 days for colorectal adenocarcinoma [62,63,64,65]. The early detection and radical excision of thin lesions offers the best chance of lowering mortality in the short-term, while prevention could play a crucial part in achieving the long-term results [66]. In accordance with these data, we believe that restricted access could have had an indirect impact on the clinical presentation and prognosis of melanoma patients in Serbia. 

Our results showed greater median Breslow thickness with a higher mitotic index that we believe is in correlation with the former. These findings coincide with other similar studies conducted during these two years [29,30,31,32,33,34,35,36,37]. Although Ricci et al. observed a significant increase in melanoma lesions thicker than 1 mm during the pandemic (0.88 vs. 1.96 mm) [30,37], we noted a 2 mm Breslow thickness depth being an important breaking point, having seen a trend of increase in pT3-4 stage melanomas during the pandemic. This is in correlation with Shannon et al., who noted an increase in median tumor depth and the proportion of T3-4 staged cases among surgical patients [31]. 

Although no age and sex distribution differences in pT staging were observed in our study, we did notice a greater Breslow thickness median in female patients as well as in patients older than 40 years of age. This former finding is in correlation with Kostner et al., who observed thicker melanomas in older females during the pandemic in Switzerland. Although these sex-related differences in melanoma are well established and go in favor of female survival rates [67,68,69], all sex-related differences in cancer diagnosis during pandemic-induced medical restrictions should be considered with concern. 

Tejera et al. published a rate of growth model based on Breslow thickness from the first moment a patient noticed the new lesion (or changes involving an existing one) to the moment of surgical excision. This model showed a shift from early toward higher pT stages, noting a 21% increase in the patient group with a one-month diagnostic delay, a 29% increase with a two-month delay, and up to a 45% increase in the three-month delay group. They reported that a three-month diagnostic delay led to a notable drop in pT1 cases (40% vs. 27%) with an accompanying doubling in pT4 cases (16% vs. 30%) as well as lower five and ten-year survival rates [70]. Ricci et al. also reported melanomas with a higher ulceration status during the pandemic (5.3% vs. 23.5%), which was not the case in our study [30]. 

Regarding the impact of pandemic-induced diagnostic delay on different cancer-type survival rates, Maringe et al. reported that in the UK, there has been an increase in the number of deaths due to breast, colorectal, esophageal, and lung cancer due to different scenarios in the past five years. For these four tumor types, an additional 3291–3621 death cases were recorded, with an estimated additional 59,204–63,229 years of life lost (YLL) [26]. Although guidelines regarding diagnosis and surgical management of melanoma during pandemic already exist [71,72,73,74], there is still a lack of sufficient data and guidelines for how to improve screening programs during these times [47,75]. 

### 4.4. Strengths and Limitations of This Study

Our study is strengthened by the fact that we used data from the largest skin cancer treatment facility in Serbia. We included a large sample and gathered data from the pandemic’s beginning down to the moment when all state-issued restrictions and regulations were lifted. 

Nevertheless, this is a single-center study of a facility in a region that was impacted early and throughout the whole pandemic and therefore has its limitations. We focused on a number of high-risk primary tumor characteristics of melanoma and did not take into account the distribution of patients with lymph node or distant metastasis across these time periods. Due to the small sample obtained during the pandemic, we could not provide a temporal subdivision of the pandemic cohort. Our facility is one of the largest of its kind in our country, with very few similar others outside of Belgrade, but nevertheless, the study could be further strengthened by including their samples as well as samples from facilities dealing with this type of pathology across the Balkan region. Although in considerably less severe clinical forms, the pandemic is still ongoing and could still impact national and global healthcare systems. Additionally, given the variety in the duration of lockdown and restrictions between countries, our findings need to be interpreted with caution. The current study is limited by the inability to study survival outcomes. Finally, our study was retrospective in nature and, thus, can be subject to the biases associated with these types of studies.

## 5. Conclusions

The COVID-19 pandemic has caused a great diagnostic and treatment delay in melanoma patients, both nationally and globally, and thus considerably disrupted all prior efforts continuously built over the past years through prevention and treatment guides. Nevertheless, this study could be an important scenario guide in the case of similar events in the future. The long-term focus regarding melanoma diagnosis and treatment should be on telemedicine and preventive measures such as national screening programs, self-examination guides, and education on the impact of melanoma on general population health.

## Figures and Tables

**Figure 1 ijerph-19-16806-f001:**
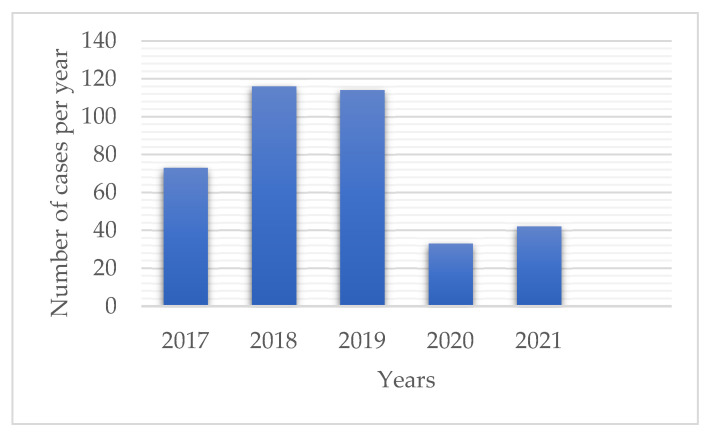
The number of diagnosed melanoma patients per year (01/JAN/17-31/DEC/21). A significant reduction in the number of diagnosed cases during the pandemic was observed (*p* = 0.036).

**Figure 2 ijerph-19-16806-f002:**
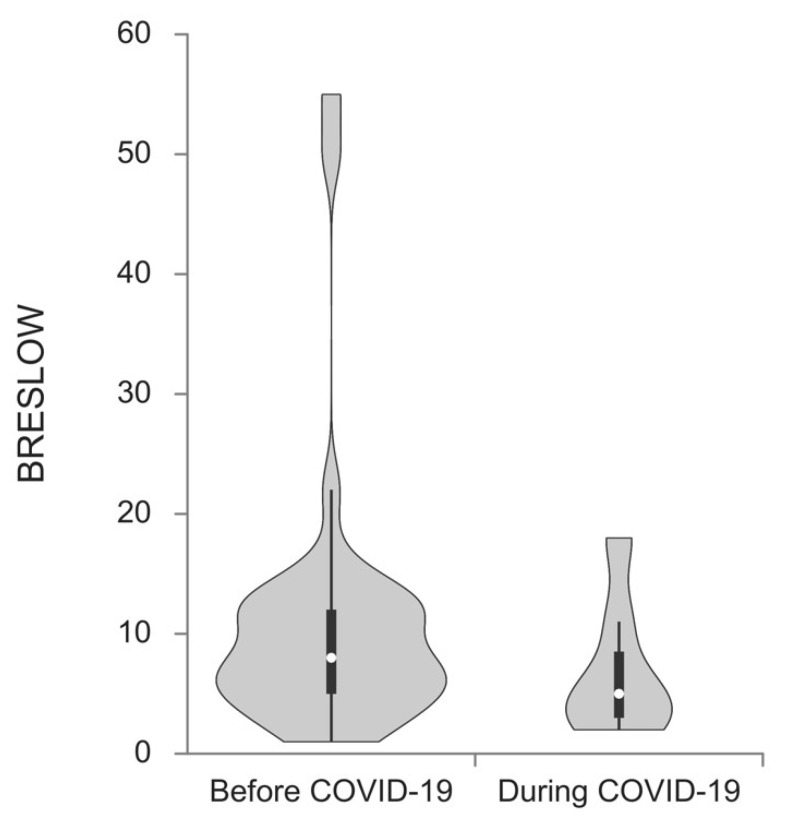
Breslow thickness median of melanoma patients diagnosed before and during the COVID-19 pandemic. Patients diagnosed during the pandemic had a higher Breslow thickness median (1.80 vs. 3.00; *p* = 0.010).

**Figure 3 ijerph-19-16806-f003:**
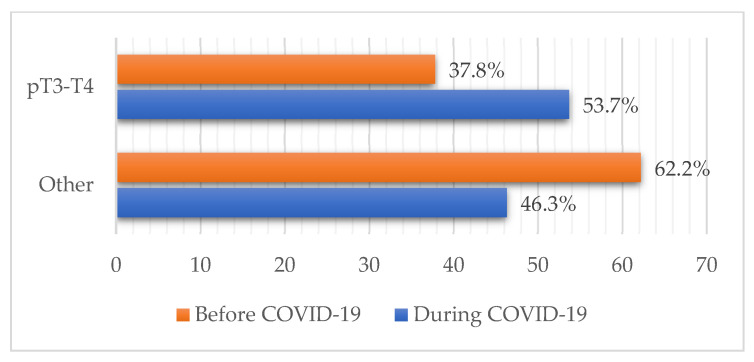
Combined pT staging of melanoma patients before and during the pandemic. An increase in melanomas thicker than 2 mm, viewed through the increase in pT3-4 staged cases was observed (37.8% vs. 53.7%; *p* = 0.026).

**Table 1 ijerph-19-16806-t001:** General characteristics of melanoma patients before and after the onset of the COVID-19 pandemic. No differences were observed between age, sex, body distribution, histologic subtype, and ulceration status in the two study groups. As shown in the table, patients diagnosed with melanoma after the onset of the pandemic had a higher Breslow thickness median. Higher Clark levels and mitotic index rate during the pandemic were also noted.

		Pre-COVID-19	COVID-19	
	Total	n = 339	n = 54	*p*
		(86.3%)	(13.7%)	Value
**Average cases per year (mean ± SD)**		101 ± 24.3	37.5 ± 6.4	0.036
**Age (mean ± SD)**		64.5 ± 15.8	65.7 ± 15.3	
**Age groups:**				0.852
**≤40 years**	39 (9.9%)	34 (10.0%)	5 (9.3%)	
**41–60 years**	104 (26.5%)	92 (27.1%)	12 (22.2%)	
**61–80 years**	191 (48.6%)	162 (47.8%)	29 (53.7%)	
**>80 years**	59 (15%)	51 (15.0%)	8 (14.8%)	
**Sex:**				0.978
**Male**	219 (55.7%)	189 (55.8%)	30 (55.6%)	
**Female**	174 (44.3%)	150 (44.2%)	24 (44.4%)	
**Body distribution:**				0.137
**Head and Neck**	81 (20.6%)	68 (20.1%)	13 (24.1%)	
**Torso**	154 (39.2%)	127 (37.5%)	27 (50.0%)	
**Upper extremities**	76 (19.3%)	70 (20.6%)	6 (11.1%)	
**Lower extremities**	82 (20.9%)	74 (21.8%)	8 (14.8%)	
**Melanoma subtype:**				0.282
**Superficial spreading**	258 (65.6%)	227 (67.0%)	31 (57.4%)	
**Lentigo maligna**	38 (9.7%)	31 (9.1%)	7 (13.0%)	
**Nodular**	73 (18.6%)	59 (17.4%)	14 (25.9%)	
**Others**	24 (6.1%)	22 (6.5%)	2 (3.7%)	
**Breslow thickness** **(median, 25th–75th percentile)**		1.80 (0.65–4.30)	3.00 (1.5–5.30)	0.010
**Breslow thickness:**				0.088
**<1 mm**	111 (28.2%)	160 (47.2%)	19 (35.2%)	
**1.01–2 mm**	57 (14.5%)	51 (15.0%)	6 (11.1%)	
**2.01–4 mm**	64 (16.3%)	51 (15.0%)	13 (24.1%)	
**>4 mm**	93 (23.7%)	77 (22.7%)	16 (29.6%)	
**pT staging:**				0.088
**In situ**	68 (17.3%)	57 (16.8%)	11 (20.4%)	
**T1**	111 (28.2%)	103 (30.4%)	8 (14.8%)	
**T2**	57 (14.5%)	51 (15.0%)	6 (11.1%)	
**T3**	64 (16.3%)	51 (15.0%)	13 (24.1%)	
**T4**	93 (23.7%)	77 (22.7%)	16 (29.6%)	
**Clark level:**				0.029
**I**	68 (17.3%)	57 (16.8%)	11 (20.4%)	
**II**	61 (15.5%)	58 (17.1%)	3 (5.6%)	
**III**	81 (20.6%)	75 (22.1%)	6 (11.1%)	
**IV**	159 (40.5%)	129 (38.1%)	30 (55.6%)	
**V**	24 (6.1%)	20 (5.9%)	4 (7.4%)	
**Mitotic index rate** **(median, 25th–75th percentile):**		2 (0–5)	5 (1–12)	<0.001
**Ulcerations present:**				0.179
**Yes**	114 (35.1%)	95 (33.7%)	19 (44.2%)	
**No**	211 (64.9%)	187 (66.3%)	24 (55.8%)	

## Data Availability

Not applicable.

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
