# Peer review of "Impact of the COVID-19 Pandemic on Melanoma Diagnosis: Increased Breslow Thickness in Primary Melanomas—A Single Center Experience"

_ijerph, 2022, doi:10.3390/ijerph192416806_

Round 1

Reviewer 1 Report

This study is not very valuable. You just proved a simple fact, that pandemic caused delays in diagnoses and hence many conditions were diagnosed in more advanced stages. You don't have to study that to know that. This also doesn't apply to melanoma only, but to many other conditions and cancers. 

Author Response

Thank you for your comment. Although the pandemic’s impact on global and national healthcare could be seen in many ways, after seeing several studies showing no differences between melanomas obtained pre and during the pandemic, we wanted to determine what was the case in Serbia, considering that no previous studies of this kind have been done in our region. We agree that the diagnostic delay’s impact applies on many other conditions and strongly believe that its long-lasting consequences could be a great area for future research.

Reviewer 2 Report

The study is important in the evaluation of the recent pandemic and supports other studies which came to similar results. In table 1 in the test of clark level it is unclear what is being tested as 3 p-values are shown. Why? Are you testing difference between specific clark levels between years, but not all or are your testing if the distribution of melanomas clark levels changed overall? 

Author Response

Thank you for your comments and suggestions. In the statistical analysis, we determined an overall change in Clark level distribution (p = 0.029) between the two groups. After reading your comments regarding the other two Clark level p-values, we realized that we had made a typographical error, for which we sincerely apologize and have since corrected.

Reviewer 3 Report

Jeremic et al. describe the impact of the COVID 19 lockdowns on dermatological healthcare.  In particular, they looked at incidence of melanoma as well as the Breslow thickness, Clark IV-V level and mitotic rate index of the lesions.  They note that in 2021 and 2022 (when restrictions were less severe) they saw an increase in pT3-T4 staging.  They discuss the impact of lockdowns on preventative care.

Minor suggestions:  Figures 1 and 3 require error bars and statistical analysis.  

Author Response

Thank you for your comments and suggestions. Variables presented in Figures 1 and 3 contain variables that cannot be presented with  error bars. The statistical analysis for both figures is shown in their legends; please see lines 166 and 205, respectively.

Reviewer 4 Report

The manuscript addresses temporal changes in the diagnosis of melanoma associated with the COVID-19 pandemic in a single large hospital in Serbia. The topic is not new, dozens of similar studies have already been published. The results are in line with the vast majority of previous studies and support the notion that the COVID-19 pandemic delayed melanoma diagnosis and negatively impacted the likelihood that melanoma cases would be diagnosed at an early stage. To my knowledge, the study is the first one contributing data on the topic from Serbia and one of the very few ones from an Eastern Europe country which makes it more relevant.

Specific remarks:

- The information on the distribution of pT staging in Table 1 and Figure 3 is redundant. This can be avoided by giving more detailed information on pT staging (i.e. the categories in situ, pT1, pT2, pT3, PT4 separately) in the Table and providing the information on the dichotomized version of the variable in the Figure. When comparing the distribution of pT staging in its detailed (non-dichotomized) version between the pre-pandemic and the pandemic period the statistical evaluation should be done with care (see next remark).

- The statistical evaluation of differences in the distribution of the Clark level between the pre-pandemic and the pandemic period is partially incorrect. The Clark level of the tumor is an ordinal variable with five categories. When comparing the distribution of such a variable between two groups, the comparison of two proportions representing the share of one category (or the combination of more than one category) of such a variable between the groups can be misleading (especially when the choice of the category to be compared is performed post hoc, meaning after the data have been looked at). The correct procedure consists of using a statistical test comparing the total distribution of the variable between the groups. If a chi-squared test is used, the ordinal information of the Clark level is ignored. If a Cochran-Armitage test is used, the ordinal information plays a role in the statistical evaluation of differences. The Cochran-Armitage test would look for a "trend" to higher Clark levels in the pandemic period compared to the pre-pandemic period. Such a trend test will have a higher statistical power than the chi-squared test when the suspected trend is correct, but the ability to detect unsuspected changes is sacrificed. Alternatively, one could think about using a Mann-Whitney test comparing the two periods with repect to differences in the distribution of the variable Clark level (which can be ranked as it is an ordinal variable).

- Figure 2 is similar to a violin plot of the distributions of Breslow's tumor thickness in the two time periods. However, a standard violin plot contains also the elements of a box plot, i.e. the median, upper/lower quartils and whiskers are also incorporated in the graphics. Figure 2 shows only the estimated probability density part of the standard violin plot. Why?

- The discussion mentions in section 4.4 some limitations of the study. The description is not complete. The low number of cases in the pandemic period resulting from the single center approach was probably the reason for avoiding a consideration of temporal subdivisions of the pandemic period. The pandemic situation was not constant over the time period defined in the study as pandemic period. A subdivision into different time periods depending on the pandemic situation in Serbia would thus have been interesting, but is limited due to the small sample size. Furthermore, the study has to assume a constant referral pattern of potential melanoma cases in the pre-pandemic and pandemic period as the basis of the validity of its analyses. For example, if more patients with suspicious skin lesions decide to consult local physisians in smaller hospitals for diagnostic work-up and avoid a large hospital serving the whole city of Belgrade and acting as as national tertiary referral center for skin cancer during the pandemic period than before, such a change could give a biased impression of temporal changes in melanoma diagnosis due to the pandemic. This potential bias is a consequence of the design of the study as a single center study. If the authors had access to national cancer registry data for melanoma they could have ruled out such a bias, but as they rely on data from a single center, they have to discuss the potential bias resulting from temporal changes in (self-)referral of patients with suspicious skin lesions for diagnostic work-up as a limitation.

- The manuscript gives proper reference to many earlier studies on the same topic. A recent study in another Eastern European country (Romania) is, however, missing and should be added (Ungureanu et al. Int. J. Environ. Res. Public Health 2022, 19, 15129. https://doi.org/10.3390).

Round 2

Reviewer 4 Report

The authors have addressed my comments appropriately in the revised version of the manuscript.